# Malaria Vaccine Introduction in Cameroon: Early Results 30 Days into Rollout

**DOI:** 10.3390/vaccines12040346

**Published:** 2024-03-22

**Authors:** Shalom Tchokfe Ndoula, Frank Mboussou, Andreas Ateke Njoh, Raoul Nembot, Simon Franky Baonga, Arnaud Njinkeu, Joseph Biey, Mohamed II Kaba, Adidja Amani, Bridget Farham, Jean-Christian Kouontchou Mimbe, Christian Armel Kouakam, Konstantin Volkmann, Crépin Hilaire Dadjo, Phanuel Habimana, Benido Impouma

**Affiliations:** 1Expanded Program on Immunization, Ministry of Public Health, Yaoundé P.O. Box 1937, Cameroon; 2World Health Organization Regional Office for Africa, Brazzaville P.O. Box 06, Congo; 3School of Global Health and Bioethics, Euclid University, Bangui P.O. Box 157, Central African Republic; 4World Health Organization Country Office, P.O. Box 155 Yaoundé, Cameroon

**Keywords:** malaria vaccine, Cameroon, new vaccine introduction, Africa

## Abstract

Cameroon introduced the malaria vaccine in its routine immunization program on 22 January 2024 in the 42 districts out of 200 that are among the most at risk of malaria. A cross-sectional analysis of the data on key vaccine events in the introduction roadmap and the vaccine uptake during the first 30 days was conducted. In addition to available gray literature related to the introduction of the malaria vaccine, data on the malaria vaccine uptake by vaccination session, collected through a digital platform, were analyzed. A total of 1893 reports were received from 22 January 2024 to 21 February 2024 from 766 health facilities (84% of overall completeness). Two regions out of ten recorded less than 80% completeness. As of 21 February 2024, 13,811 children had received the first dose of the malaria vaccine, including 7124 girls (51.6%) and 6687 boys (48.4%). In total, 36% of the children were vaccinated through outreach sessions, while 61.5% were vaccinated through sessions in fixed posts. The overall monthly immunization coverage with the first dose was 37%. Early results have shown positive attitudes towards and acceptance of malaria vaccines. Suboptimal completeness of data reporting and a low coverage highlight persistent gaps and challenges in the vaccine rollout.

## 1. Introduction

Malaria is a tropical and subtropical mosquito-borne parasitic disease that continues to cause unacceptably high levels of illness and deaths in sub-Saharan Africa, despite progress being made in disease control [1]. The disease is transmitted by female *Anopheles* mosquitoes that breed in freshwater bodies in warm and humid environments [2]. Severe and fatal malaria is predominantly caused by *Plasmodium falciparum*, which accounts for more than 90% of the world’s malaria mortality [3]. In 2022, 249 million malaria cases were recorded in 85 malaria-endemic countries, providing a case incidence of 58 cases per 1000 population at risk [4]. The available preventive tools for malaria include vector control measures, chemoprophylaxis, and preventive chemotherapies [5].

Vaccination has been recently added to the malaria prevention package, following the prequalification by the World Health Organization (WHO) of two vaccines: RTS/AS01 (GSK Biologicals, Rixensart, Belgium), approved in 2022 [6], and R21 (University of Oxford, Oxford’s Jenner Institute Laboratories, Oxford, UK), approved in 2023 [7]. These two vaccines are recommended as additional tools for preventing *Plasmodium falciparum* malaria in children living in areas with moderate to high malaria transmission. RTS, S/AS01 was piloted as part of the Malaria Vaccine Implementation Programme (MVIP) in three African countries (Ghana, Kenya, and Malawi). Nearly 2 million children have received the vaccine through this initiative since 2019 [8]. These pilot programs demonstrated the RTS,S/AS01 vaccine’s safety and effectiveness, and the use of the vaccine led to a substantial reduction in severe malaria cases and a significant decrease in child deaths, even in areas where other preventive measures were present [8], such as long-lasting insecticidal nets, intermittent preventive treatment and seasonal malaria chemoprevention. R21 has recently reached the primary one-year endpoint in a pivotal large-scale phase III clinical trial, including 4800 children across Burkina Faso, Kenya, Mali, and Tanzania [9].

Cameroon is one of the 11 countries with a high malaria burden [4]. The disease is widespread and endemic and is the leading cause of morbidity and mortality in children under five years of age [10,11]. In 2022, Cameroon reported 5.7 million suspected malaria cases, including 3.3 million confirmed cases and 2481 related deaths [4,5]. The malaria-preventive interventions in Cameroon include vector control measures such as long-lasting insecticidal nets, intermittent preventive treatment administered to pregnant women since 2015, seasonal malaria chemoprevention (implemented since 2017 in the Far North and North Regions), and perennial malaria chemoprevention rolled out in 2022. In addition, malaria cases are managed with medications as per the current WHO guidelines.

Following the prequalification of RTS,S/AS01 and its successful pilot rollout in three African countries, the Cameroon government decided to introduce the malaria vaccine into its routine immunization program and as part of the malaria control package. The country applied for RTS,S/AS01 allocations through the Gavi Vaccine Introduction grant and received the first shipment of 331,200 doses (out of the 701,211 doses required in 2024) on 21 November 2023 [10]. The Cameroon Expanded Programme on Immunization (EPI) developed a roadmap towards the effective rollout of the malaria vaccine, including indicators for periodically monitoring national and district readiness. As a result of subnational disease burden and vaccination impact analyses, 42 health districts that are among the most at risk for malaria were selected for the introduction of the vaccine across the country’s ten regions. On 22 January 2024, Cameroon kicked off the malaria vaccine rollout.

This paper summarizes the introduction process, data completeness from health facilities, and preliminary vaccine uptake results one month in to the vaccine rollout.

## 2. Materials and Methods

A cross-sectional analysis was conducted using data on key vaccine events of the malaria vaccine introduction (MVI) roadmap and vaccine uptake during the first 30 days of the rollout in Cameroon.

### 2.1. Study Settings

The health system in Cameroon is divided into ten regions and 200 health districts. Forty-two districts that are among the most at risk for malaria were selected for the MVI following subnational tailoring analysis [12]. Table 1 presents the number and percentage of districts chosen for the MVI by region. Each health district comprises health areas containing one or more health facilities ensuring vaccination services. All the health facilities providing routine vaccinations were selected for the MVI. The health facilities had to carry out combined, fixed, outreach, and mobile vaccination sessions to reach all the eligible children in their catchment areas.

Figure 1 shows the geographical distribution of the districts selected for the MVI by region.

### 2.2. Eligibility and Inclusion and Exclusion Criteria


As per the national plan for RTS,S/AS01 vaccine introduction, the schedule comprises four doses that are administered to children at 6, 7, 9, and 24 months of age. The first malaria dose is co-administered with vitamin A.Children born before July 2023, i.e., aged 7 months or above in January 2024, were not eligible for malaria vaccines.Data from health facilities outside the 42 selected districts and any record of the second or third malaria vaccine dose were considered errors and excluded accordingly.All public and private health facilities providing routine vaccinations in the 42 districts were allowed to deliver malaria vaccines.


### 2.3. Data Sources

Available gray literature related to the MVI in Cameroon, including minutes of the National Immunization Technical Advisory Group (NITAG) and the MVI Technical Working Group (TWG) meetings, the national plan for MVI, and official statements related to MVI, were reviewed to build a dataset of events of the MVI roadmap in Cameroon.

Data from the national database on routine immunization in the 42 districts that were selected for the MVI were also analyzed. In these districts, data on routine immunization were collected by vaccination session using a digital form deployed in IASO, an open-source geo-structured data collection platform developed by Bluesquare [13], a global health information systems and data management company supporting the Ministry of Health in Cameroon. Data on the number of children who were vaccinated by a health facility and antigen, including malaria vaccine doses, were disaggregated by gender and by vaccination strategy (fixed, outreach, and mobile).

### 2.4. Data Analysis

Data on key events were analyzed to draw a timeline of key events from the NITAG recommendation to the vaccine rollout kick-off.

Using the dataset on vaccine uptake from the IASO platform, duplicate records based on the region name, district name, health area name, health facility name, vaccination strategy, and date of reporting were systematically removed using a script that keeps only the first record.

The following parameters were computed:Reporting completeness: number of reports received from health facilities (by vaccination session), reported during a specified period (one month in this study) and divided by the total number of expected reports in the 42 selected districts and multiplied by 100. The number of expected reports was calculated by multiplying the number of health facilities by the average number of vaccination sessions per health facility and month.Immunization coverage: number of eligible children who have received the first dose of malaria vaccine divided by the total number of eligible children at the assessment date, multiplied by 100.Number of unvaccinated children: total eligible children at the assessment date minus the number of children who received the first malaria vaccine dose.Percentage of girls vaccinated: number of girls vaccinated divided by the total number of children vaccinated, multiplied by 100.Percentage of boys vaccinated: number of boys vaccinated divided by the total number of children vaccinated, multiplied by 100.Percentage of children vaccinated in fixed sessions: number of children vaccinated during fixed sessions divided the total number of children vaccinated, multiplied by 100.Percentage of children vaccinated in outreach sessions: number of children vaccinated during outreach sessions divided by the total number of children vaccinated, multiplied by 100.Percentage of children vaccinated in mobile sessions: number of children vaccinated during mobile sessions divided by the total number of children vaccinated, multiplied by 100.

All data analyses and visualizations were performed using R software, version 4.3.1 [14].

This study used data abstracted from the Ministry of Health database and did not require ethical clearance.

## 3. Results

### 3.1. Timeline of Key Events Related to the MVI

Figure 2 shows the timeline of key events, from the recommendation of RTS,S/AS01 by the NITAG to the effective start of vaccine rollout. The MVI was kicked off 383 days (12.8 months) after the recommendation of RTS,S/AS01 by the NITAG and 62 days (2 months) after the reception of the first shipment of malaria vaccines.

### 3.2. Completeness of Data Reporting from Health Facilities

Based on the number of vaccination sessions planned in each of the 802 health facilities that were selected for the MVI, 2240 reports were expected every month, including 1893 reports received from 22 January to 21 February 2024 from 766 health facilities, providing 84% overall completeness.

Figure 3 presents the distribution of the completeness in terms of the number of reports received over the expected number by region.

Two regions recorded less than 80% completeness: Centre (52%) and North-West (69%). Two other regions received more reports than expected: North (112%) and South (117%). This is because some health facilities carried out additional vaccination sessions, especially during the first week of MVI.

### 3.3. Vaccination Sessions Carried Out by Health Facilities

A total of 3026 vaccination sessions were carried out from 22 January 2024 to 21 February 2024 by 766 health facilities, providing an average of 3.9 sessions per health facility. The median number of vaccination sessions carried out by the health facilities was 1 [range: 1–7] in West, 1 [range: 1–8] in North-West, 1 [range: 1–15] in Centre, 2 [range: 1–10] in Adamaoua, 2 [range: 1–11] in East, 2 [range: 1–12] in Littoral, 2 [range: 1–21] in South, 3 [range: 1–9] in North, and 3 [range 1–37] in Far North. Figure 4 shows the distribution of the number of vaccination sessions carried out during the first month of MVI by the 766 health facilities that submitted reports on the malaria vaccine uptake.

Overall, of the 766 health facilities that reported on the malaria vaccine uptake, 555 conducted fewer than four vaccination sessions during the first month of introduction (72%). A total of 211 health facilities carried out at least one vaccine session per week (38%).

Of the 3026 vaccination sessions carried out, 2064 were fixed (68.2%), 915 were outreach (30.2%), 31 were mobile (1.0%), and 16 were not specified (0.6%). The median percentage of outreach sessions was 13.6%, ranging from 1.9% in the South-West Region to 46.7% in the Far-North Region and 48.7% in the North Region.

### 3.4. Children Vaccinated with the First Dose of RTS,S/AS01

As of 21 February 2024, 13,811 children had received the first dose of the malaria vaccine, including 7124 girls (51.6%) and 6687 boys (48.4%). The number of children who were vaccinated during the ISO week four, corresponding to the first week of the MVI and the third week of the month of January, was 2827 across the 42 districts. This number increased by 53% in ISO week five (fourth week of January) compared to the previous week (*n* = 4320) before decreasing by 41% in ISO week six (first week of February) (*n* = 2570) and increasing by 16% in ISO week seven (second week of February 2024) (*n* = 2975), but this was still below the level of ISO week five. Figure 5 presents the distribution of the number of children who were vaccinated with the first dose of the malaria vaccine by ISO week and region and at the national level from 22 January to 21 February 2024.

Of the 13,811 children who were vaccinated with RTS,S/AS01, 8497 were vaccinated through fixed sessions (61.5%), 4976 through outreach sessions (36.0%), and 282 through mobile sessions (2.0%); and the strategy was not specified for 56 children (0.5%). Figure 6 presents the distribution of children who were vaccinated from 22 January 2024 to 21 February 2024 by strategy and region.

The North, Far-North, and East Regions recorded the highest number of children being vaccinated and are the regions with the highest proportion of children who were vaccinated through outreach sessions: 51.7%, 50.1%, and 36.9%, respectively.

### 3.5. Co-Administration of RTS,S/AS01 and Vitamin A

The first dose of RTS,S/AS01 is administered simultaneously with vitamin A. Overall, the number of administered vitamin A doses was lower than those of RTS,S/AS01: 10,988 versus 13,811. In all regions, the number of administered RTS,S/AS01 doses was higher than the number of vitamin A doses, except in the South Region (Figure 7).

### 3.6. Vaccination Coverage

The monthly immunization coverage with the first dose of the malaria vaccine was 37% (13,811 children vaccinated out of the 37,208 in the target population) from 22 January 2024 to 21 February 2024; this means that 23,397 children in the target population did not receive the malaria vaccine (63%). Figure 8 presents the distribution of children who were vaccinated and unvaccinated with the first dose of RTS,S/AS01 and the immunization coverage with the first dose of RTS,S/AS01 by region as of 21 February 2024. North (52%), Littoral (44%), and East (43%) were the regions with the highest immunization coverage. The Far-North Region recorded a coverage below the national average despite being the region with the second highest absolute number of children who were vaccinated with the first dose of RTS,S/AS01.

Figure 9 shows the geographical distribution of the immunization coverage with the first dose of RTS,S/AS01 by district. We found that two districts, including one in the East (Moloundou) and one in the Centre Region (Soa), recorded over 80% coverage with the first dose of the malaria vaccine (5% of the 42 selected districts), while 12 districts recorded less than 30% coverage (28%), 21 districts between 30% and 49% coverage (50%), and seven districts between 50% and 79% (17%). The two districts surpassing 80% coverage were among the five with the lowest target population.

## 4. Discussion

Cameroon is the first country to introduce the malaria vaccine into its routine immunization program outside the pilot program that was implemented in three countries (Ghana, Kenya, and Malawi). This introduction occurred after the ones for the Human Papillomavirus (HPV) vaccines in 2020 and COVID-19 vaccines in 2021 that resulted in a high vaccine hesitancy and low coverage [15,16,17,18]. Exposure to misinformation and a low risk perception were the main drivers of vaccine hesitancy in the case of COVID-19 and HPV vaccine in Cameroon. Despite a high level of political commitment, the spread of misinformation suggesting that the malaria vaccine was dangerous and ineffective following the receipt of the first shipment led to the initial launch of MVI, scheduled for 12 December 2024, being postponed. In a commentary published in September 2023, Titanji et al. [19] warned that the rollout of the malaria vaccine coincided with the prevailing challenge of vaccine misinformation and hesitancy, amplified during the COVID-19 pandemic and that assumptions about vaccine acceptance might be faulty. The postponement of the launch gave the Cameroon Ministry of Health time to intensify community-based risk communication involving community leaders and health workers rather than using the usual mass media communication. This targeted risk communication and a high malaria risk perception contributed to the relatively successful launch of MVI in Cameroon, as shown by the figures provided in this paper. Kimbi et al. [20] found that 92% of pregnant women and caregivers of under-five in Buea health district, Cameroon, had the correct perception of malaria vaccines.

A total of 13,811 children received the first dose of the malaria vaccine during the first month after its rollout in Cameroon. This number may be underestimated, as the completeness of expected reports from health facilities was about 84%. In addition to integrating the malaria vaccine into a routine immunization data system based on end-of-month reporting of aggregated data, Cameroon piloted a new system of reporting routine immunization data by vaccination session [21]. This system uses an electronic form, deployed in IASO and developed by Bluesquare [13]. Such suboptimal completeness in data reporting, especially for a new and additional reporting system, could, in part, stem from the fact that many frontline immunization staff in Cameroon are commonly overburdened with multiple data-related responsibilities that compete with their clinical tasks and affect their data collection practices [22]. Temeslow Mamo [23] from the Tony Blair Institute for Global Change identified three cross-cutting data management challenges related to the use of digital platforms for vaccine rollouts, including weak connectivity and infrastructure (lack of data collection equipment and poor connectivity in remote districts), poor data management systems (multiple reporting tools, data not integrated into a single system or platform, and a lack of interoperability across various systems), anda shortage of a trained workforce. Through its EPI, the Cameroon Ministry of Health must intensify supportive supervision to identify and address issues preventing completeness in malaria vaccine uptake data reporting.

In this paper, regions that carried out the highest proportion of outreach vaccination sessions, such as the North and Far-North Regions, recorded the highest absolute number of children being vaccinated with the first dose of the malaria vaccine. Akoh et al. [24], from an assessment of the immunization service delivery in one of the largest health districts in the West Region of Cameroon, reported a low proportion of health facilities (4.8%) that organized outreach sessions within the 3 months before the study. Geographical accessibility to vaccination services is known as one of the main drivers of vaccination coverage in Cameroon, so outreach vaccination sessions play a pivotal role in reducing immunization inequities. For Streefland et al. [25], the routine provision of vaccinations using outreach strategies in addition to static facilities has become the backbone of sustainable vaccination systems in developing countries. Investing more in outreach sessions as part of routine immunization could improve the performance of MVI in Cameroon.

The coverage of the monthly target population with the first dose of the malaria vaccine at the end of the first month of MVI in Cameroon was low (37%). Even though it is too early to compare the Cameroonian experience with the three pilot programs, the vaccination coverage for the first dose in 2020 (the first year of implementation) was 88%, 71%, and 69% in Malawi, Ghana, and Kenya, respectively [26]. In Cameroon, the fact that the number of children who received the first dose of the malaria vaccine in the first 30 days was greater than the number of those who received vitamin A in most regions (the two products being administered at the same time in the immunization schedule), means that there is little or no refusal of malaria vaccines in health facilities. The low coverage in Cameroon may result either from overestimating the target population or under-reporting through the new system being piloted in the 42 districts that were selected for MVI. Intensified supportive supervisions could help to rule out the fact that a significant portion of eligible children who reported in health facilities for other vaccines are still to receive the malaria vaccines, as one of the causes of low coverage in Cameroon. Indeed, the first dose of malaria vaccine in Cameroon is not co-administered with any other vaccine of the routine immunization schedule, except vitamin A which usually records low uptake. Carrying out supportive supervision to identify challenges early was among the main lessons learned from the three pilot programs [27]. Visits to health facilities by experts from the EPI to witness vaccination in action and provide on-site training to health workers contribute to identifying operational gaps and ways to address them [27].

### Limitations

The data used in this paper were reported by health facilities from the digital platform for reporting immunization data by vaccination session. No data quality audit has been conducted to assess the consistency of the data between paper-based registries and the digital tool.

The completeness of reports received out of expected was about 84%. This may have led to an under-estimation of the number of children who were vaccinated with the first dose of the malaria vaccine.

Duplicate records based on the region name, district name, health area name, health facility name, vaccination strategy, and date of reporting were systematically removed using a script that kept only the first record. This could have led to removing the right record and keeping the wrong one.

This study focuses on quantitative data on vaccine uptake. Qualitative data on community perceptions and misinformation that may lead to vaccine hesitancy were not collected. This has prevented the study from analyzing contextual factors associated with the low vaccine uptake.

The findings of this report should, therefore, be interpreted with these limitations in mind.

## 5. Conclusions

The recent introduction of the malaria vaccine into routine immunization in Cameroon marks a decisive turning point in malaria control. RTS,S/AS01 malaria vaccines have now been administered to eligible children in the most affected districts for malaria in Cameroon for one month. The early results have shown positive attitudes towards and acceptance of malaria vaccines, evidenced by the fact that the number of administered malaria doses was greater than that of doses of vitamin A in most regions. However, the suboptimal completeness of reported data and the low coverage highlight persistent gaps and challenges in the vaccine rollout. Additional training for health workers may be needed to enhance the data collection and reporting mechanisms and ensure higher data completeness rates. National authorities should contemplate undertaking qualitative research to explore factors associated with low coverage, including those related to vaccine hesitancy and the effectiveness of risk communication strategies, as well as geographical and socio-economic contexts.

There is also an urgent need to conduct a post-introduction evaluation to rapidly identify operational gaps and recommend corrective measures, since the country will soon start providing the second dose of the vaccine. Health facilities’ compliance with the guidance on vaccination for eligible children reporting late, the effectiveness of tracking the access to malaria vaccine for eligible children reporting to health facilities for other vaccines, and the drivers of delayed or no reporting should be among the focus areas in such an evaluation. An early post-introduction evaluation will also help to draw lessons for other countries in the African region that have planned to introduce the malaria vaccine in 2024 and 2025. In addition, planning for long-term follow-up studies to access the vaccine’s safety and impact on malaria morbidity and mortality, as well as the incremental benefits of integrating the malaria vaccine into routine immunization, should be carried out.

## Figures and Tables

**Figure 1 vaccines-12-00346-f001:**
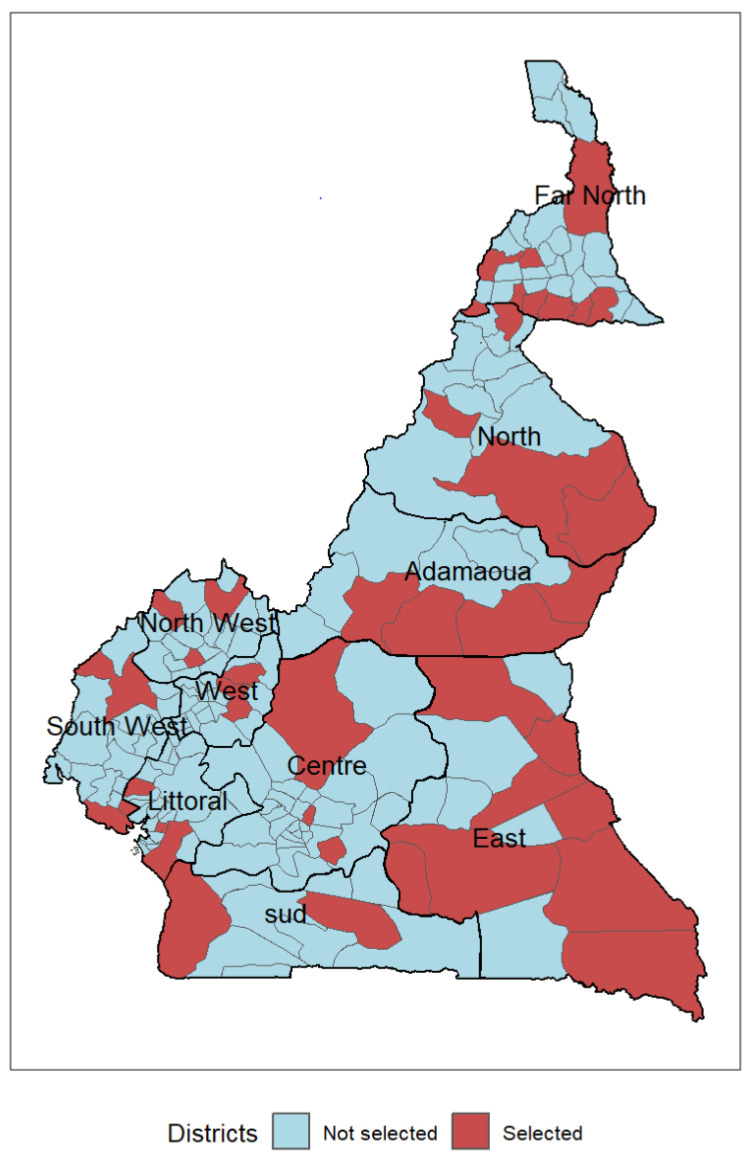
Geographical distribution of districts most at risk of malaria selected for the malaria vaccine introduction in Cameroon, 2024.

**Figure 2 vaccines-12-00346-f002:**
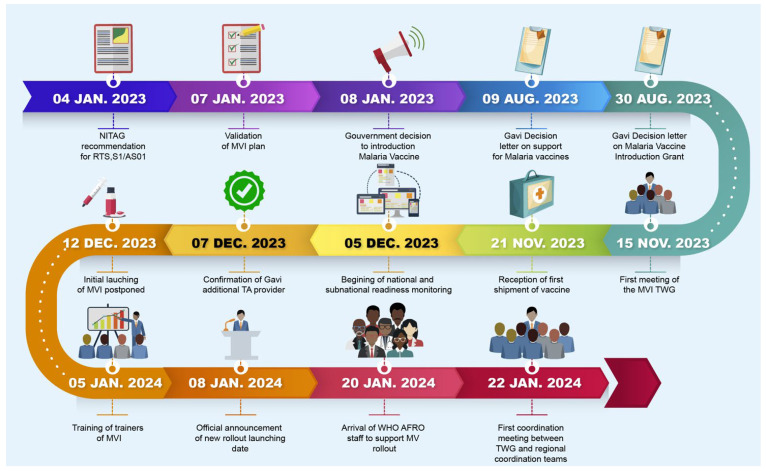
Timeline of key events related to the malaria vaccine introduction in Cameroon. MV: malaria vaccine; WHO AFRO: World Health Organization Regional Office for Africa; MVI: malaria vaccine introduction; NITAG: National Immunization Technical Advisory Group; TWG: Technical Working Group.

**Figure 3 vaccines-12-00346-f003:**
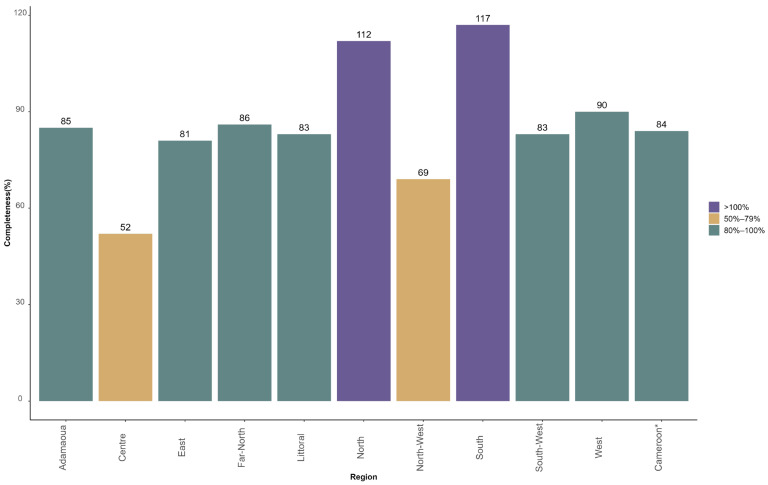
Percentage of reports on malaria vaccine uptake received from health facilities over expected number from 22 January to 21 February 2024 (data from 42 districts of Cameroon selected for malaria vaccine introduction). * Cameroon corresponds to 42 districts selected for malaria vaccine introduction.

**Figure 4 vaccines-12-00346-f004:**
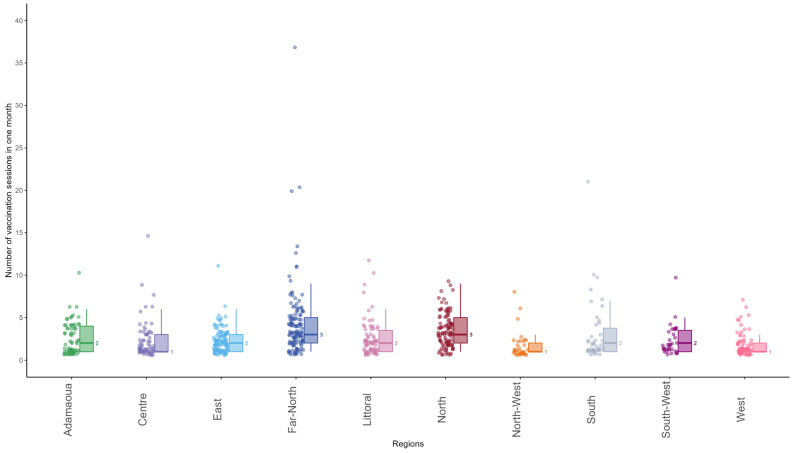
Distribution of number of vaccination sessions carried out from 22 January to 21 February 2024 by 766 health facilities selected for malaria vaccine rollout, by region in Cameroon.

**Figure 5 vaccines-12-00346-f005:**
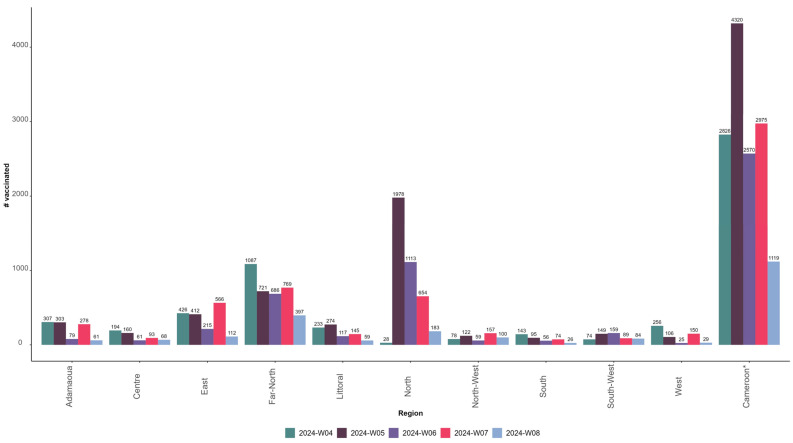
The number of children vaccinated with the first dose of the malaria vaccine from 22 January to 21 February 2024 by ISO week and region in Cameroon. * Cameroon corresponds to the 42 districts selected for the malaria vaccine introduction.

**Figure 6 vaccines-12-00346-f006:**
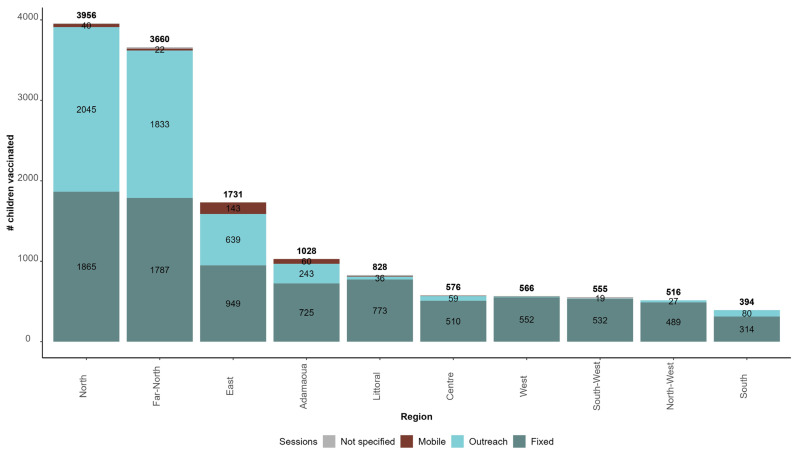
Number of children vaccinated with malaria vaccine by vaccination strategy and region in Cameroon from 22 January 2024 to 21 February 2024.

**Figure 7 vaccines-12-00346-f007:**
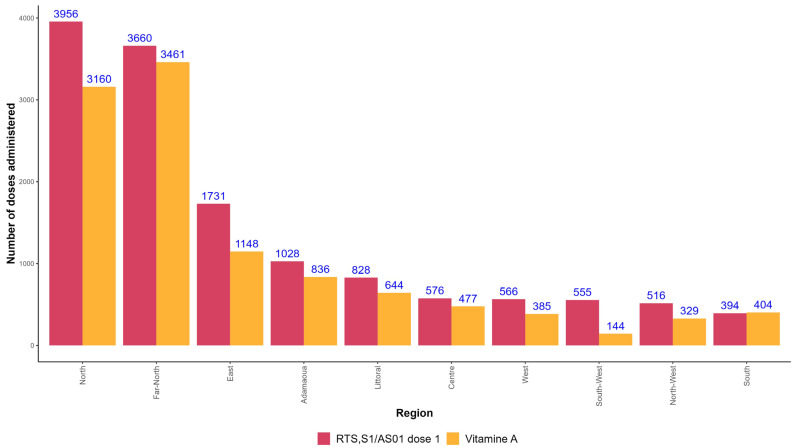
Doses of RTS,S/AS01 and vitamin A administered to children in 42 districts selected for malaria vaccine introduction in Cameroon from 22 January 2024 to 21 February 2024.

**Figure 8 vaccines-12-00346-f008:**
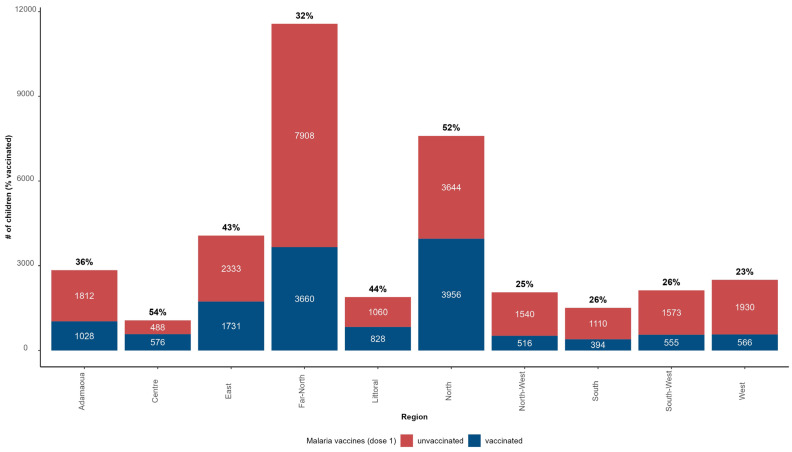
Number of children who were vaccinated or unvaccinated with RTS,S/AS01 and related immunization coverage in 42 districts in Cameroon (data from 22 January 2024 to 21 February 2024).

**Figure 9 vaccines-12-00346-f009:**
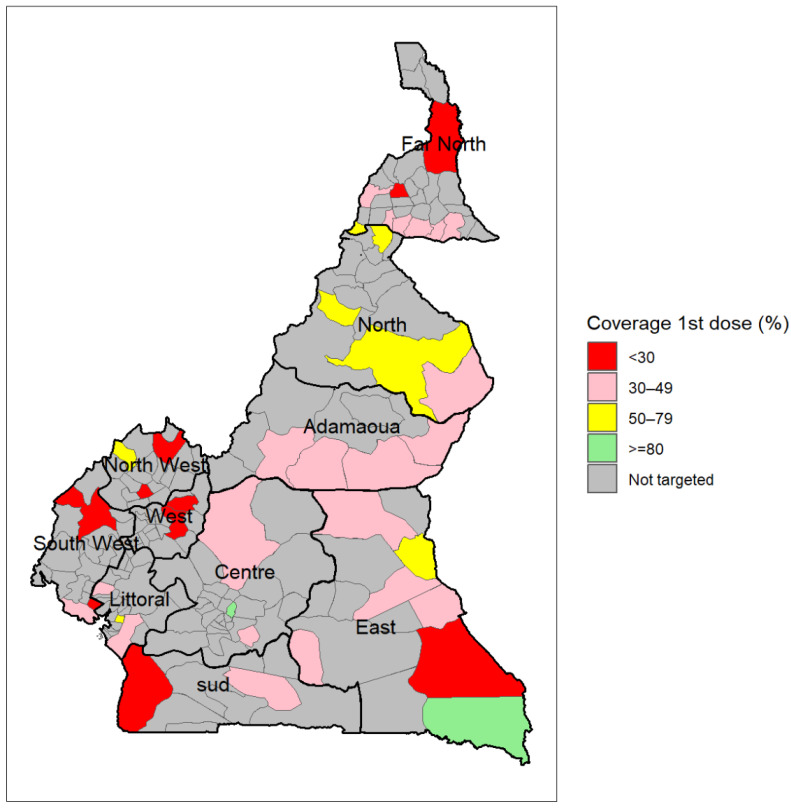
Immunization coverage with the first dose of RTS,S/AS01 by district in Cameroon (data as of 12 February 2024).

**Table 1 vaccines-12-00346-t001:** Number of districts most at risk of malaria and health facilities selected for the malaria vaccine introduction in Cameroon, 2024.

Region	Total Number of Districts	Number of Districts Selected	% Districts Selected	Number of Health Facilities
Adamaoua	11	4	40	65
Centre	32	3	9	125
East	15	8	53	115
Far North	32	9	28	125
Littoral	24	3	12	82
North	15	4	27	74
North-West	20	3	15	44
South	12	3	30	36
South-West	19	3	16	43
West	20	2	10	93
**Cameroon**	**200**	**42**	**20**	**802**

## Data Availability

The data presented in this study are available from the corresponding author upon request.

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
