# Peer review of "Malaria Vaccine Introduction in Cameroon: Early Results 30 Days into Rollout"

_vaccines, 2024, doi:10.3390/vaccines12040346_

Round 1
Reviewer 1 Report
Comments and Suggestions for Authors
The paper provides an insightful analysis of the initial phase of the malaria vaccine introduction in Cameroon. This detailed review will cover the study's strengths and drawbacks, followed by recommendations for improvement.
Strengths:
1. The study employs a cross-sectional analysis of data from 766 health facilities, providing a robust dataset on vaccine uptake and implementation effectiveness within the first 30 days of rollout.
2. By reporting vaccination data disaggregated by gender and age, the study offers valuable insights into vaccine coverage among different demographic groups.
3. The differentiation between outreach and fixed vaccination sessions provides a nuanced understanding of the vaccination strategy's effectiveness.
4. Given the ongoing global efforts to combat malaria, the study's focus on the early results of vaccine rollout in a high-burden country is timely. It contributes valuable insights for policy and practice.
Drawbacks:
1. The study reports an 85% completeness rate in data reporting from health facilities, which may lead to underestimation of vaccination coverage and impact.
2. Although the paper mentions the vaccine rollout in 42 districts, there is insufficient detail on the geographic distribution of vaccination sessions, particularly the challenges in hard-to-reach areas.
3. The study largely relies on quantitative data, which may overlook the contextual factors affecting vaccine uptake, such as community perceptions, misinformation, and vaccine hesitancy.
4. There is a missed opportunity to compare these early rollout results with similar vaccine introduction efforts in other countries or with other vaccines introduced in Cameroon.
Recommendations:
1. Enhance data collection and reporting mechanisms to ensure higher completeness rates. This could involve additional training for healthcare workers on data entry and timely and complete reporting.
2. To complement the quantitative findings, undertake qualitative research to explore the reasons behind vaccine hesitancy, acceptance, and the effectiveness of different communication strategies.
3. Provide a more detailed analysis of the vaccine rollout across different regions, focusing on the challenges and successes in diverse geographic and socio-economic contexts.
4. Compare the malaria vaccine rollout with other vaccine introductions in Cameroon or other countries. This could offer insights into best practices and strategies to improve future vaccine rollout efforts.
5. Address the sustainability of the vaccine introduction effort by planning long-term follow-up studies to assess vaccine impact, booster needs, and integration into the routine immunization schedule.
Author Response
- Strengths
1.1. The study employs a cross-sectional analysis of data from 766 health facilities, providing a robust dataset on vaccine uptake and implementation effectiveness within the first 30 days of rollout.
1.2. By reporting vaccination data disaggregated by gender and age, the study offers valuable insights into vaccine coverage among different demographic groups.
1.3. The differentiation between outreach and fixed vaccination sessions provides a nuanced understanding of the vaccination strategy's effectiveness.
- 4. Given the ongoing global efforts to combat malaria, the study's focus on the early results of vaccine rollout in a high-burden country is timely. It contributes valuable insights for policy and practice.
Response: Thank you having highlighted the strengths of our study. This is much appreciated.
- Drawbacks:
2.1. The study reports an 85% completeness rate in data reporting from health facilities, which may lead to underestimation of vaccination coverage and impact.
Response: Thank you for this comment. The suboptimal completeness has been considered as a limitation of this study (see line 330-331).
2.2. Although the paper mentions the vaccine rollout in 42 districts, there is insufficient detail on the geographic distribution of vaccination sessions, particularly the challenges in hard-to-reach areas.
Response: Thank you for this comment. We have added the following sentence to address this comment (see lines 92-96):” Each health district comprises of health areas containing one or more health facilities ensuring vaccination services. All the health facilities providing routine vaccinations were selected for the MVI. The health facilities had to combined fixed, outreach and mobile vaccination sessions to reach all the eligible children in their catchment areas.”
2.3. The study largely relies on quantitative data, which may overlook the contextual factors affecting vaccine uptake, such as community perceptions, misinformation, and vaccine hesitancy.
Response: Thank you for this comment. We have added this drawback among the study limitations. The following sentence has been added (see lines 335-337):” This study focuses on quantitative data on vaccine uptake. Qualitative data on community perceptions and misinformation leading to vaccine hesitancy were not collected. This has prevented the study from analyzing contextual factors associated with low vaccine uptake.”
- 4. There is a missed opportunity to compare these early rollout results with similar vaccine introduction efforts in other countries or with other vaccines introduced in Cameroon.
Response: Thank you for this comment. In the discussion section, the fact that the malaria vaccine introduction after the introduction of Human Papillomavirus (HPV) vaccine in 2020 and COVID-19 vaccines in 2021 in Cameroon was discussed. The contribution of malaria risk perception to reducing vaccine hesitancy compared to HPV and COVID-19 vaccines was highlighted and supported by few references (see lines 261-277).
- Recommendations
3.1. Enhance data collection and reporting mechanisms to ensure higher completeness rates. This could involve additional training for healthcare workers on data entry and timely and complete reporting.
3.2. To complement the quantitative findings, undertake qualitative research to explore the reasons behind vaccine hesitancy, acceptance, and the effectiveness of different communication strategies.
3.3. Provide a more detailed analysis of the vaccine rollout across different regions, focusing on the challenges and successes in diverse geographic and socio-economic contexts.
3.4. Compare the malaria vaccine rollout with other vaccine introductions in Cameroon or other countries. This could offer insights into best practices and strategies to improve future vaccine rollout efforts.
3.5. Address the sustainability of the vaccine introduction effort by planning long-term follow-up studies to assess vaccine impact, booster needs, and integration into the routine immunization schedule.
Response: Thank you for these suggestions. We have added these recommendations in the conclusion section (see lines 347-351 and lines 359-362).
Reviewer 2 Report
Comments and Suggestions for Authors
Thank you for sharing your article on early findings following the introduction of the malaria vaccine in selected, endemic regions of Cameroon. The article reads very well. Here just few comments may help to improve the article.
L45 & L46: Not sure to what "their" relates to; please clarify.
L47: What other preventive measures were present when conducting the vaccine programmes?
L52: Probably "under five" years; please add the appropriate unite.
L52-53: Please clarify how malaria cases can be suspected and confirmed at the same time; please re-phrase. What criteria did suspected versus confirmed cases have to meet? How was/is malaria confirmed across Cameroon?
L108: Antigen other than Plasmodium?
L111: Calculations are presented for the first malaria vaccine dose. Is it anticipated to generate those figures for the other/upcoming time points too?
Figure 3 and 5: I am wondering whether the category "Cameroon" is a cumulative figure? Please clarify.
L217: It may be good for future readers to mention the administration of Vitamin A in the materials & methods section. Seeing the data in the result section is somewhat surprising.
Author Response
L45 & L46: Not sure to what "their" relates to; please clarify.
Response: Thank you for this comment. This has been clarified (see line 53-54).
L47: What other preventive measures were present when conducting the vaccine programmes?
Response: Thanks for this comment. We have provided the list of other preventive measures (see line 56-58).
L52: Probably "under five" years; please add the appropriate unite.
Response: Thanks for this suggestion. This has been addressed (see line 63).
L52-53: Please clarify how malaria cases can be suspected and confirmed at the same time; please re-phrase. What criteria did suspected versus confirmed cases have to meet? How was/is malaria confirmed across Cameroon?
Response: Thanks for this comment. This is how data are reported in the Malaria Word Report (Reference 4). Countries report to WHO suspected cases based on a case definition included in national guidance and parasitologically confirmed by microscopy or Rapid Diagnostic Test. We have added reference 5 which provides the definition of malaria suspected and confirmed cases.
L108: Antigen other than Plasmodium?
Response: Thanks for this comment. We cannot find a sentence referring to antigen other than plasmodium.
L111: Calculations are presented for the first malaria vaccine dose. Is it anticipated to generate those figures for the other/upcoming time points too?
Response: Thanks for this comment. Given that only the first dose was administered during the first month, we did not find it relevant to provide the calculation of parameters related to other/upcoming doses.
Figure 3 and 5: I am wondering whether the category "Cameroon" is a cumulative figure? Please clarify.
Response: Thanks for this comment. We have updated the figures and add a footnote on Cameroon.
L217: It may be good for future readers to mention the administration of Vitamin A in the materials & methods section. Seeing the data in the result section is somewhat surprising.
Response: Thanks for this suggestion. This has been addressed (line 105-106).